# Foot-and-Mouth Disease Virus-like Particles Produced in *E. coli* as Potential Antigens for a Novel Vaccine

**DOI:** 10.3390/vetsci12060539

**Published:** 2025-06-02

**Authors:** Sang-Cheol Yu, In-Kyu Lee, Hyun-Seok Kong, Sung-Ho Shin, Sung-Yoon Hwang, Yu-Jin Ahn, Jong-Hyeon Park, Bong-Yoon Kim, Young-Cheon Song

**Affiliations:** 1Watson RnD Sharing Co., Ltd., #1003, 282, Sunhwagung-ro, Namyangju-si 12106, Gyeonggi-do, Republic of Korea; watsonrnd10@gmail.com (S.-C.Y.); watsonrnd12@gmail.com (I.-K.L.); watsonrnd27@gmail.com (Y.-J.A.); 2Department of Animal Science, Sahmyook University, Seoul 01795, Republic of Korea; hahj@syu.ac.kr; 3Center for Foot-and-Mouth Disease Vaccine Research, Animal and Plant Quarantine Agency, 177 Hyeoksin 8-ro, Gimcheon City 39660, Gyeongsangbuk-do, Republic of Korea; ikarus121@korea.kr (S.-H.S.); hsy8592@korea.kr (S.-Y.H.); parkjhvet@korea.kr (J.-H.P.); 4College of Pharmacy, Sahmyook University, Seoul 01795, Republic of Korea

**Keywords:** foot-and-mouth disease virus (FMDV), virus-like particles (VLPs), *E. coli* expression system, vaccine immunogenicity, animal challenge model

## Abstract

Foot-and-mouth disease (FMD) is a highly contagious viral infection that causes severe economic losses in livestock industries worldwide. Although conventional vaccines made from inactivated viruses are widely used, they have several drawbacks, including the need for high-level biosafety facilities, challenges in large-scale production, and reduced effectiveness against new virus strains. In this study, we developed and tested virus-like particle (VLP) vaccines for three important FMD virus serotypes (O-, A-, and Asia1) using a bacterial expression system in *Escherichia coli*. The resulting VLPs closely resembled natural viruses in structure and successfully triggered strong immune responses in mice and pigs. Importantly, the vaccines provided dose-dependent protection, with high levels of immunity achieved even at lower doses. These results suggest that VLP vaccines could be a safer, more scalable, and cost-effective alternative to traditional FMD vaccines. This approach could help improve global access to effective FMD vaccines, benefiting animal health, food security, and agricultural economies.

## 1. Introduction

Foot-and-mouth disease (FMD) is an acute, highly contagious, and economically devastating disease that primarily affects cloven-hoofed animals such as pigs, cattle, and goats, and is caused by infection with FMD virus (FMDV) [1]. Due to its rapid spread (up to 250 km by air) and the difficulties of prevention after an outbreak has occurred, infected animals are often culled to prevent further infection, resulting in significant economic losses [1,2]. Currently, in spite of efforts to prevent outbreaks worldwide, recent global surveillance of FMD suggests that it is continuously endemic in several parts of Asia, large parts of Africa, and the Middle East [3]. Vaccination is the most effective tool for preventing FMD, and current vaccines are mostly based on inactivated FMDV antigens. Although conventional inactivated FMDV vaccines have been effectively reducing the number of FMDV outbreaks in many parts of the world where the disease is endemic, these vaccines require biosafety level 3 (BSL-3) facilities for virus cultivation and have high production costs. They also have limited efficacy against antigenically variable strains and issues related to antigenic stability and supply chain logistics [4,5,6].

Virus-like particles (VLPs) are self-assembled macromolecules composed of viral capsid proteins but lacking genetic materials. They mimic the morphology of native virions and are considered one of the most promising platforms for next-generation vaccines due to their high safety and immunogenicity profiles [7,8]. Previously, FMDV VLPs produced in *E. coli* and mammalian systems through co-expression of the FMDV capsid P1-2A precursor protein and 3C protease showed high protection potency in swine and cattle [9]. Notably, it has been demonstrated that co-expression of FMDV P1 and a modified 3C protease (L127P) in *E. coli* enables intracellular cleavage and spontaneous assembly of VLPs, confirming the feasibility of the in vivo particle formation in the *E. coli* expression system [10]. These VLPs elicit neutralizing antibodies and cellular immunity by exposing multiple conformational epitopes that are similar to native FMDV particles [11,12]. Additionally, research on the stability and immunogenicity of empty FMDV particles has highlighted the importance of maintaining their structural integrity to elicit a strong immune response [13].

However, most prior research has focused on single FMDV serotypes or relied on eukaryotic expression systems such as insect or mammalian cells, which are more complex and costly. To date, few studies have systematically evaluated the structural integrity and immunogenicity of multivalent FMDV VLPs produced in a scalable bacterial system such as *E. coli*. In addition, comprehensive comparative evaluations using both mice and swine models, particularly including PD_50_-based challenge assessments, are limited or lacking.

In this study, we produced highly expressed self-assembled FMDV VLPs of serotypes O-, A-, and Asia1 in the *E. coli* system. We then examined and confirmed their VLP formation and structural integrity via biochemical and biophysical analysis. In addition, we evaluated the immunological efficacy of each FMDV VLP vaccine using ELISA and virus neutralization testing (VNT) in mice and pigs. Taken together, our results suggest that *E. coli*-derived FMDV VLPs are promising next-generation vaccine candidates that can address the limitations associated with conventional inactivated FMDV vaccines.

## 2. Materials and Methods

### 2.1. Plasmid Construction

To construct the gene assembly vectors for the production of FMDV VLP antigens, the genes encoding the P1-2A region and the 3C protease region of foot-and-mouth disease virus were synthesized and amplified via PCR using specific primers (Table 1 and Figure 1). The sequences of foot-and-mouth disease viruses used were O PanAsia2 (O PA2, O/TUR/5/2009, GenBank accession No. KP202878.1) for serotype O, A Yeoncheon (A YC, A/SKR/Yeoncheon/2017, GenBank accession No. KY766148.1) for serotype A, and Asia1 Shamir (As1/Shamir/89, GenBank accession No. JF739177.1) for serotype Asia1-type. The generated genes were cloned into the expression vectors pET-duet and pACY-duet.

### 2.2. Preparation of FMDV VLP Antigen

To express the FMDV VLP in *E. coli* cells, gene constructs were transformed into *E. coli* BL21 (DE3) (Enzynomics Co., Ltd., Daejeon, Republic of Korea) and selected by reference to Amp, Kan, and Chl resistance. To express the scale-up, protein expression was performed in a fermenter (MARADO-Double, BIOCNS, Ltd., Daejeon, Republic of Korea), and the culture medium was prepared as previously described [14]. The fermentation process was started at 37 °C with stirring at 200 rpm, increasing to 350 rpm after 5 h, and the air flow rate was set to 0.25 v.v.m. Induction of protein expression was achieved using autoinduction for 24 h.

Upon completion of expression, *E. coli* cells were lysed using a microfluidizer (LM20, Microfluidics™, Worcester, UK) at 10,000 psi using 50 mM Tris-HCl. Cell debris was removed via centrifugation and the supernatant containing the target protein was collected. After filtration of the collected FMDV VLPs using a 0.22 μm filter system (HYUNDAI MICRO, Seoul, Republic of Korea), the VLPs were purified using 6000 and 8000 precipitations of polyethylene glycol (PEG). To purify the FMDV VLPs from filtered cell lysis supernatant, the O-type and A-type FMDV VLP expression cell lysis supernatants were incubated in a reaction mixture of 7.5% PEG6000 (Sigma-Aldrich, St. Louis, MO, USA) containing 0.5 M NaCl, while the Asia1-type FMDV VLP was incubated in a reaction mixture of 10% PEG8000 (Sigma-Aldrich, USA) containing 0.5 M NaCl overnight. The precipitated protein was then collected via centrifugation and the purified target protein was eluted using a buffer containing 20 mM Tris-HCl and 300 mM KCl.

### 2.3. Characterization and Measurement of FMDV VLPs

The purified VLPs were quantified using the BCA assay (Thermo Scientific, Waltham, MA, USA) and analyzed using SDS-PAGE (non-reducing and reducing) electrophoresis and Western blotting. Non-reducing conditions were performed using guinea pig polyclonal anti-FMDV antibodies produced by immunizing guinea pigs with a commercially available inactivated FMDV vaccine (Boehringer Ingelheim, Bracknell, UK) at a 1:1000 dilution as the primary antibody. HRP-conjugated goat anti-guinea pig IgG antibodies (Abcam, Cambridge, UK) were used as the secondary antibody at a 1:3000 dilution. Reducing conditions were performed using FMDV-VP1 polyclonal antibody received from Dr. Kim of the Rural Development Administration (RDA), which was used as the primary antibody at a dilution of 1:1000. HRP-conjugated goat rabbit IgG antibodies (Abcam, UK) were used as the secondary antibody at a 1:3000 dilution. Blots were treated in 5% (*w*/*v*) skim milk for 30 min and then washed three times for 5 min in 1x PBS-T buffer. Primary antibodies were treated overnight at 4 °C and the membranes were washed three times for 10 min each with 1x PBS-T buffer. Anti-lgG HRP secondary antibodies (Abcam, UK) were applied to the membrane for 1 h at room temperature and washed three times for 10 min with 1x PBS-T. After its reaction, the HRP was reacted with an enhanced chemiluminescence (ECL) solution (Thermo Fisher Scientific, Waltham, MA, USA) and then confirmed with Chemidac (FluorChem E System, Bio-techne, Minneapolis, MN, USA).

To observe the morphology of the purified VLP, we requested the Korea Basic Science Institute (KBSI) to take a transmission electron microscope photograph, and the purified VLP was separately adsorbed on a carbon-coated copper grid and visualized with a transmission electron microscope (JEM1010, JEOL Ltd., Akishima, Japan).

### 2.4. Purification of VLPs Using Sucrose Gradient Centrifugation

To purify the VLPs, 1 mL of concentrated PEG-precipitated FMDV VLPs were added to the top of 11 mL of a 15–45% (*w*/*v*) sucrose density gradient and centrifuged at 100,000× *g* for 4 h at 4 °C using a swinging-bucket rotor (SW41Ti, Beckman Coulter, Brea, CA, USA). The supernatant was collected and the peaks of each fraction were measured using HPLC, the absorbances at 280 nm of the gradient sample were confirmed via HPLC analysis. HLPC conditions were in accordance with previously described methods [15,16].

### 2.5. Vaccine Preparation and Mouse Immunization

Animal experiments were conducted in strict accordance with the recommendations of the Animal and Plant Quarantine Agency (APQA) guide for the care and use of laboratory animals. All animal procedures were approved by the Institutional Animal Care and Use Committee (IACUC) of APQA, Republic of Korea (Approval No. 2024-860). All possible efforts were made to minimize the suffering of the animals. Female C57BL/6 mice of 7 weeks of age (KOSABIO, Seongnam-si, Republic of Korea) were intramuscularly immunized. The experimental vaccine consisting of FMD virus-like particle antigen (30 μg/dose) was formulated with ISA 206 VG (Seppic, Courbevoie, France, water-oil-water double oil emulsion, W/O/W type), Quil-A^®^ adjuvant (InvivoGen, San Diego, CA, USA), and 10% aluminum hydroxide gel adjuvant (InvivoGen, USA). For each dose, ISA 206 VG was added at 50% (*v*/*v*), Quil-A^®^ at 100 µg per dose, and aluminum hydroxide gel at 10% (*v*/*v*). The vaccine used as a control was compared with a commercial vaccine, i.e., a foot-and-mouth disease vaccine from Boehringer Ingelheim (UK). Serum was collected at 21 and 42 days post-vaccination (dpv), and a booster vaccination was administered at 28 dpv.

### 2.6. Detection of FMDV-VLP Specific Antibodies in Mouse Sera

The presence of antibodies to FMD-VLP-specific antibody levels in the peripheral blood of mice was determined using the PrioCHECK™ FMDV Type A Antibody ELISA Kit (Thermo Fisher Scientific, USA) for serotype A-type, the PrioCHECK™ FMDV Type O Antibody ELISA Kit (Thermo Fisher Scientific, USA) for serotype O-type, and the PrioCHECK™ FMDV Type Asia1 Antibody ELISA Kit (Thermo Fisher Scientific, USA) for serotype Asia1-type, following the manufacturer’s instructions. Animals with inhibition ratio (PI) values ≥50% were considered antibody positive.

### 2.7. Neutralizing Antibody Titer Assay in Mouse Sera

The virus neutralization (VN) test was performed as described in the OIE terrestrial manual [17]. LFBK cells were seeded into 96-well plates and cultured in a 5% CO_2_ incubator at 37 °C until 80% confluency. Serum samples collected from mice 21 and 42 dpv were heat-inactivated at 56 °C for 30 min. Serially diluted 50 µL serum samples were reacted with 50 µL of FMDV (100 TCID_50_) and incubated at 37 °C for 1 h. Subsequently, after incubation at 37 °C for 1 h, 50 µL of LFBK cells (0.5 × 10^6^ cells/mL) were added to each well, and a blank negative control was included and incubated in 5% CO_2_ for 3 days. The cytopathic effect (CPE) was assessed to determine neutralizing antibody titers, expressed as the log10 of the reciprocal antibody dilution required to neutralize 100 TCID_50_ of virus. The virus neutralization test (VNT) used FMDV strains representing the following: A/SKR/Yeoncheon/2017 (serotype A), O/VET/2013 (serotype O), and or As1/Shamir/89 (serotype Asia1). Data are presented as mean ± standard deviation (SD) and statistical significance was determined at *p* < 0.05.

### 2.8. Challenge Study in Immunized Mice

C57BL/6 mice (n = 5 per group, KOSABIO, Republic of Korea) were vaccinated with ISA 206 (W/O/W type) mixed with different doses of VLP antigen (0, 1/10, 1/40, 1/160, and 1/640 groups of 30 µg/dose antigen), Quil-A^®^ adjuvant (InvivoGen, USA), and 10% aluminum hydroxide gel adjuvant. Seven days post-vaccination (dpv), each mouse was challenged intraperitoneally with 100 LD_50_ of FMDV strains representing the following: A/SKR/Yeoncheon/2017 (serotype A), O/VET/2013 (serotype O), or As1/Shamir/89 (serotype Asia1). Survival and body weight changes were monitored up to 7 days post-challenge (dpc).

### 2.9. Challenge Study in Immunized Swine

Swine (n = 5 per group, XPBio, Republic of Korea) were vaccinated with ISA 206 (W/O/W type) mixed with different doses of VLP antigen (0, 1/1, 1/4, and 1/16 groups of 300 µg/dose antigen), Quil-A^®^ adjuvant (InvivoGen, USA), and 10% aluminum hydroxide gel adjuvant. For each dose, ISA 206 VG was added at 50% (*v*/*v*), Quil-A^®^ at 1 mg per dose, and aluminum hydroxide gel at 10% (*v*/*v*). At 28 days post-vaccination (dpv), the swine were challenged with 100 LD_50_ of FMDV strains (O/Boeun/2017 (serotype O) or Asia1/Shamir/89 (serotype Asia1)) via intradermal injection into the foot. Clinical observations were made daily and clinical scores were calculated based on specific criteria, including body temperature; lameness; hoof and foot vesicles; and snout, lip, and tongue vesicles.

### 2.10. Control Groups

To assess the immunogenicity and protective efficacy of the FMDV VLP vaccines, both positive and negative control groups were included in all animal experiments. The positive control group was administered a commercially available inactivated FMD vaccine (Boehringer Ingelheim, UK) according to the same immunization schedule used for the experimental groups. Serum samples collected from these animals were utilized as reference controls in ELISA and virus neutralization tests (VNTs) to facilitate comparative analysis of immune responses elicited by the VLP formulations. The negative control group received a mock formulation containing no antigen, consisting of Tris-KCl buffer (20 mM Tris-HCl, 3000 mM KCl, pH 7.5) combined with the same adjuvant mixture used in the VLP vaccines: ISA 206 VG (50% *v*/*v*), Quil-A^®^ (10 µg/mouse or 100 µg/swine), and aluminum hydroxide gel (10% *v*/*v*). These animals underwent the same procedures as the vaccinated groups, including immunization, blood sampling, and viral challenge. Sera from the negative controls were included in serological analyses and PD_50_ calculations. All animals in the control groups were maintained under the same environmental conditions and sampling timelines as the experimental groups to ensure procedural consistency and reduce experimental bias.

### 2.11. Statistical Analysis

Statistical relationships between inoculated animals and groups were determined using the Kruskal–Wallis test followed by Dunn’s multiple comparisons tests and the log-rank test, performed using GraphPad Prism (Ver 5.0; GraphPad Software, Boston, MA, USA) with significance set at *p* < 0.05.

## 3. Results

### 3.1. Expression and Purification of FMD VLPs

To produce the FMD VLPs, two copies were made of each serotype of FMDV P1-2A/3C (L127P) genes (O-, A-, and Asia1-type), followed by cloning into multiple cloning sites of pET-duet and pACY-duet expression vectors (Figure 2A,B). Each plasmid for FMDV VLPs was separately transformed into *E. coli* BL21 (DE3) cells, cultured in a fermenter using an appropriate culture medium, and disrupted using a microfluidizer. The FMDV VLPs were analyzed via Western blotting under non-reducing SDS-PAGE conditions. As shown in Figure 2C,D, FMD VLPs had very high molecular weights, i.e., larger than 245 kDa, and this band was almost the same as that of the purified inactivated FMD virus antigens produced via sucrose gradient centrifugation. Western blot analysis under reducing conditions further confirmed the expression of VP1 (27 kDa) for each serotype (Figure 2D). This result suggests that recombinant FMDV VLPs are self-assembled in the *E. coli* expression system.

### 3.2. Structural Characteristics of FMDV VLPs

The purified FMD VLPs were further examined using transmission electron microscopy (TEM) for morphological and biochemical analysis. As expected, TEM analysis clearly showed each serotype of the FMDV VLPs (O-type, A-type, and Asia1-type), measuring 25~30 nm in diameter, with typical morphology and size (Figure 3A–C). Subsequently, FMDV VLPs were concentrated and purified using sucrose gradient centrifugation, with fractions 7~8 showing enriched VLPs compared with other high and low molecules (Figure 3D). Moreover, Western blotting and HPLC analysis using purified and concentrated VLPs revealed that a strong VLP-enriched band (Figure 3E) and peaks were observed at 14.5 min, as observed for the purified inactivated FMDV antigens (Figure 3F). Additionally, TEM analysis of the sucrose-purified VLPs reaffirmed the uniform morphology and expected particle size distribution (25–30 nm) (Appendix A). These findings indicate that the VLPs retained structural integrity and closely resembled FMD virions.

### 3.3. Evaluation of the Efficacy of FMD VLPs in Mice After Vaccination

Each serotype of the FMDV VLP vaccine formulation contained 30 µg of purified recombinant FMDV VLP antigens. Female C57BL/6 mice (n = 5 per group) received a primary immunization followed by a booster at 28 dpv, mimicking a two-dose vaccination regimen. Blood samples were collected at 0, 21, and 42 dpv to monitor the development of humoral immune responses (Figure 3A).

Serological analysis was performed using structural protein (SP)-specific enzyme-linked immunosorbent assays (ELISAs) for each serotype (A, O, and Asia1) and virus neutralization tests (VNTs) to quantify functional antibody titers capable of inhibiting viral replication (Figure 4B–E and Figure 5A–C). To evaluate efficacy, mice were challenged with 100 LD_50_ of homologous FMDV strains A/SKR/Yeoncheon/2017 (serotype A), O/VET/2013 (serotype O), and Asia1/Shamir/89 (serotype Asia1) at 21 and 42 dpv.

Kinetic changes in percent inhibition (PI) values at 0, 21, and 42 dpv were presented for each serotype to visualize the humoral immune response over time (Figure 4B). The temporal trends allowed for clear comparison between vaccine groups, demonstrating a time-dependent increase in PI values following immunization.

The A-type FMDV VLP (30 μg/dose) group exhibited strong immunogenicity, as evidenced by a protection index (PI) exceeding 50% in the SP-A ELISA at 42 days post-vaccination (dpv) (Figure 4C), a threshold generally regarded as indicative of protective immunity. Correspondingly, virus VNT results show a marked increase in neutralizing antibody titers, rising from an average of 1:112 at 21 dpv to an average of 1:676 at 42 dpv, indicative of a robust booster response (Figure 5A).

Similarly, mice immunized with the O-type FMDV VLP (30 μg/dose) vaccine displayed consistent immunogenic responses, with PI values exceeding 50% at 21 and 42 dpv as measured using SP-O ELISA (Figure 4D). Neutralizing antibody titers reached an average of 1:143 and an average of 1:767 at 21 and 42 dpv, respectively, reflecting rapid seroconversion and a strong anamnestic response (Figure 5B).

The Asia1-type FMDV VLP (30 μg/dose) group also showed favorable serological responses. The SP-Asia1 ELISA revealed PI values that remained above 50% throughout the experimental period (Figure 4E), and VNTs demonstrated a progressive increase in neutralizing antibody titers from an average of 1:47 at 21 dpv to an average of 1:295 at 42 dpv (Figure 5C). Although the absolute titers were relatively lower than those elicited by the A and O serotype VLPs, they remained above the protective threshold and were superior to those induced by conventional inactivated vaccines. Although lower than titers for A and O serotypes, the values exceeded the protective threshold and were higher than those induced by conventional inactivated vaccines (Table 1).

### 3.4. Virus Challenge of Mice After Vaccination

Mice (n = 5 per group) were immunized with the following graded doses of each vaccine formulation: 1/10 dose (30 μg), 1/40 dose (7.5 μg), 1/160 dose (1.875 μg), and 1/640 dose (0.468 μg) in a 200 μL injection volume. Seven days post-immunization, all groups were challenged with 100 LD_50_ of the homologous FMDV strain. Serotype-specific viruses were used for the challenge, as follows: A/SKR/Yeoncheon/2017 (serotype A), O/VET/2013 (serotype O), and Asia1/Shamir/89 (serotype Asia1) (Figure 6A).

In the group immunized with the A-type FMDV VLP vaccine, 100% survival was observed at the 1/10, 1/40, and 1/160 doses, whereas 60% of mice survived at the 1/640 dose (Figure 6B). The surviving animals gradually regained body weight, indicative of recovery from infection (Figure 6C). The protective efficacy (PD_50_) of the A-type FMDV VLP vaccine was calculated to be 73.5.

For the group immunized with the O-type FMDV VLP vaccine, 100% survival was recorded at the 1/10 and 1/160 doses. Survival rates of 80% and 20% were observed at the 1/40 and 1/640 doses, respectively (Figure 6D). Similar to the A-type FMDV VLP, surviving animals exhibited progressive body weight recovery (Figure 6E). The PD_50_ of the O-type FMDV VLP vaccine was calculated to be 32.

Finally, in the Asia1-type FMDV VLP vaccine group, 100% survival was achieved at the 1/10 and 1/40 doses, while 80% and 60% survival rates were noted at the 1/160 and 1/640 doses, respectively (Figure 6F). Consistent with other groups, surviving animals exhibited gradual weight recovery (Figure 6G). The PD_50_ of the Asia1-type FMDV VLP vaccine was calculated to be 55.7.

### 3.5. Virus Challenge of Swine After Vaccination

Swine (n = 5 per group) were immunized with the following graded doses of each vaccine formulation: 1 dose (300 μg), 1/4 dose (75 μg), and 1/16 dose (18.75 μg) in a 2 mL injection volume. At 28 days post-immunization, all groups were challenged with 100 LD_50_ of the homologous FMDV strain. Serotype-specific viruses were used for the challenge, as follows: O/Boeun/2017 (serotype O) and Asia1/Shamir/89 (serotype Asia1) (Figure 7A).

In the group immunized with the O-type FMDV VLP vaccine, a 100% protection rate was observed at the 1/4 dose, whereas 80% and 20% of swine were protected at the full and 1/16 doses, respectively (Figure 7B). The protective efficacy (PD_50_) of the O-type FMDV VLP vaccine was calculated to be 10.6. In the group immunized with the Asia1-type FMDV VLP vaccine, a 100% protection rate was recorded at both the full and 1/4 doses, while a protection rate of 75% was observed at the 1/16 dose (Figure 7C). The PD_50_ of the Asia1-type FMDV VLP vaccine was calculated to be 22.6.

Among the three vaccine candidates, the A-type VLP exhibited the highest protective efficacy in mice, while the Asia1-type VLP showed the highest cross-species protection, achieving the greatest PD_50_ value in swine (Table 2). These results underscore the serotype-specific immunogenicity and protective potential of the developed VLP formulations, supporting their application in FMD control programs.

## 4. Discussion

### 4.1. Immunogenicity and Protective Efficacy of FMDV VLPs

In this study, foot-and-mouth disease virus-like particles for serotypes O, A, and Asia1 were successfully generated using recombinant plasmids expressed in the *E. coli* system. Each purified FMDV VLP exhibited a uniform morphology with a diameter of approximately 25~35 nm, as observed via transmission electron microscopy. Western blot analysis confirmed the presence of structural proteins, indicating correct antigenic conformation and structural integrity. This finding is consistent with previous studies showing that co-expression of FMDV P1 and a modified 3C protease (L127P) in *E. coli* enables intracellular cleavage and in vivo self-assembly of virus-like particles [10]. Their study provided direct evidence that *E. coli* expression systems can support in vivo assembly of morphologically authentic VLPs without the need for in vitro refolding or molecular chaperones. These findings are consistent with previous reports demonstrating that VLPs can structurally and antigenically resemble native virions, thereby serving as potent immunogens capable of eliciting strong immune responses [12,18,19,20].

The immunogenicity potential of the FMD VLPs was assessed in a murine model. Mice administered 30 µg of VLPs exhibited robust seroconversion, as confirmed by structural protein (SP) ELISA and virus neutralization (VN) assays. In the A-type FMDV VLP-immunized group, the percentage inhibition (PI) exceeded 50% at 42 dpv in SP-A ELISA, with VN titers of 1:112 and 1:676 detected at 21 and 42 dpv, respectively. In the O-type FMDV VLP-immunized group, PI exceeded 50% after 21 dpv in SP-O ELISA, with VN titers of 1:143 and 1:767 detected at 21 and 42 dpv, respectively. Similarly, in the Asia1-type FMDV VLP-immunized group, PI exceeded 50% after 21 dpv in SP-Asia1 ELISA, with VN titers of 1:47 and 1:295 at 21 and 42 dpv, respectively. These findings are in agreement with previous studies demonstrating that VLP-based vaccines effectively induce strong humoral immune responses [21].

Moreover, the antigen dose used in this study (30 µg per mouse) is comparable to, or lower than, those reported in other *E. coli*–expressed FMDV VLP studies, such as the 50 µg dose employed in a previous investigation [18]. Similarly, a vaccine based on an HBcAg-derived virus-like particle (VLP) platform presenting HIV-1 Gag epitopes required a minimum antigen dose of 100 µg, formulated with complete Freund’s adjuvant (CFA), to achieve immunogenicity levels comparable to those of conventional vaccines [22]. These results suggest that the FMDV VLPs developed in the present study may exhibit superior structural stability and antigenic presentation, enabling effective immune activation at a lower antigen dose. This antigen-sparing effect is particularly advantageous for cost-effective, large-scale production of veterinary vaccines [23].

Following challenge with 100 LD_50_ of each O-, A-, and Asia1-type FMDV virus, i.e., O-Vet, A-Yeoncheon, and Asia1-Shamir, the vaccinated mice displayed dose-dependent protection. The A-type FMDV VLP-immunized group exhibited complete protection (100% survival) at the 1/10, 1/40, and 1/160 doses, with a PD_50_ of 73.5. Similarly, the O-type FMDV VLP-immunized group achieved 100% survival at both 1/10 and 1/160 doses, with a PD_50_ of 32. The Asia1-type FMDV VLP-immunized group achieved 100% survival at the 1/10 and 1/40 doses, with a PD_50_ of 55.7.

In addition, following challenge with 100 LD_50_ of each of the O- and Asia1-type FMDV viruses, i.e., O-BE and Asia1-Shamir, the vaccinated swine displayed dose-dependent protection. In the O-type FMDV VLP-vaccinated group, complete protection (100% survival) was observed at the 1/4 dose, with a calculated PD_50_ value of 10.6. Similarly, in the Asia1-type FMDV VLP-vaccinated group, 100% survival was achieved at both the full and 1/4 doses, with a PD_50_ value of 22.6. Survival rates declined with further dose reductions across all groups. These results highlight the dose-dependent nature of FMDV VLP vaccines and are consistent with previous studies demonstrating the efficacy of FMDV VLP-based vaccines in conferring protective immunity [11,12,18].

Notably, differences in protective efficacy among serotypes may, in part, be explained by inherent replication kinetics and immunogenic characteristics. For example, serotype O FMDV has been reported to exhibit faster replication both in vitro and in vivo, which may shorten the window for effective immune priming and consequently require higher antigen doses to achieve sufficient protection [24,25,26]. This suggests that, even when VLPs maintain structural consistency, their immunological performance may vary depending on serotype-specific viral dynamics. Furthermore, the potential for persistent infection and carrier state formation in host animals may warrant additional considerations in the development of future vaccine formulations [27,28,29].

### 4.2. Broader Implications, Limitations, and Future Perspectives

The enhanced immunogenicity and protective efficacy observed with the FMDV VLP vaccine are likely attributable to the structural stability and optimized antigenic presentation of the assembled particles. Specifically, the A- and Asia1-type FMDV VLPs exhibited higher PD_50_ values relative to the O-type, indicating superior immunological potency. These findings are consistent with previous reports highlighting the pivotal role of VLP structural integrity in driving robust immune activation [30]. The pronounced protective effect observed in this study further underscores the significance of epitope stabilization and rational vaccine design strategies in maximizing immunogenic outcomes. Beyond their efficacy in this model, the structural and immunological advantages of the FMDV VLP platform suggest broader applicability to other viral diseases. This is further supported by the successful implementation of VLP-based vaccines targeting human papillomavirus (HPV), hepatitis B virus (HBV), and SARS-CoV-2 [31,32,33].

Moreover, the production strategy employed in this study underscores the feasibility and scalability of bacterial expression platforms for large-scale vaccine manufacturing. In contrast to traditional inactivated virus vaccines that necessitate biosafety level 3 (BSL-3) containment due to the use of infectious agents, VLPs can be produced without handling live viruses, thereby significantly reducing biosafety concerns and overall production costs. Previous studies have demonstrated that VLP-based vaccines are not only cost-effective but also amenable to high-throughput production, making them suitable candidates for widespread immunization programs [34,35].

Comparative evaluation demonstrated that the FMD VLP vaccines developed in this study induced immunogenicity and protective efficacy comparable to or surpassing those of licensed inactivated vaccines. However, several key challenges must be addressed before these candidates can be used in practical applications. These include ensuring the scalability of production, maintaining antigen consistency during large-scale manufacturing, ensuring regulatory compliance, and confirming safety and efficacy through field trials in target livestock species [11,18,36,37]. These results highlight the potential of VLP-based vaccines as a safer and more efficacious alternative, owing to their capacity to elicit potent immune responses without the biohazard risks associated with live virus manipulation during production. Accumulating evidence suggests that VLPs can be structurally engineered to present multiple antigenic epitopes, thereby providing a versatile platform for the development of multivalent vaccines targeting diverse serotypes [38,39].

Despite the encouraging outcomes observed, several limitations remain that warrant further investigation. First, although the mouse model offers preliminary insights into the immunogenicity and protective potential of the VLP vaccines, validation in target livestock species, such as cattle and swine, is essential to assess their efficacy and safety under practical field conditions [40,41]. Second, the durability of immune responses, including the induction of long-term immunity and immunological memory, has not yet been characterized and should be explored in subsequent studies [42]. Moreover, optimization of the production platform is required to enhance scalability and cost-efficiency, aligning with the demands of large-scale implementation in global FMD control efforts. Finally, in light of recent advances in adjuvant technology and formulation science, the incorporation of suitable adjuvants and delivery strategies to augment the immunogenicity of VLP-based vaccines deserves further attention [21,43].

Recent advances in *E. coli*-based expression systems, including the application of SUMO fusion tags and dual plasmid designs, have shown promise in improving protein yield and proper VLP assembly [11,44]. Previous studies have shown that the use of SUMO fusion tags enhances the protective efficacy of VLP vaccines in pigs and cattle, and VLP production in Pichia pastoris has also been reported to yield high immunogenicity and scalability for various viral targets [11,45,46]. These studies underscore the need for continued innovation in microbial expression platforms for vaccine development.

In summary, this study shows the successful development of virus-like particle (VLP) vaccines targeting FMDV serotypes O, A, and Asia1, demonstrating potent immunogenicity and robust protective efficacy in a murine model. These results support the continued advancement of VLP-based platforms as a promising alternative for the prevention and control of foot-and-mouth disease. Given their favorable safety profile, production scalability, and strong immune-stimulating capacity, VLP vaccines offer considerable potential as next-generation candidates in the field of veterinary vaccinology.

## 5. Conclusions

This study demonstrates the successful development of foot-and-mouth disease virus-like particles (FMDV VLPs) targeting serotypes O, A, and Asia1 using a recombinant *Escherichia coli* expression system. The resulting VLPs exhibited proper structural and antigenic integrity, effectively mimicking native virions, and induced robust, serotype-specific immune responses in a murine model. Protective efficacy analysis revealed clear dose-dependent protection, with PD_50_ values comparable to or exceeding those of conventional inactivated vaccines. Specifically, the use of a bacterial expression platform enabled scalable, rapid, and cost-effective production of vaccine antigens without the need for high-level biosafety containment or live virus handling. Collectively, these findings support *E. coli*-derived FMDV VLPs as a promising alternative to traditional inactivated vaccines, offering significant advantages in safety, production efficiency, and economic feasibility. Further validation in target livestock species and large-scale production trials will be essential to advance this vaccine platform toward practical field application and regulatory approval.

## Figures and Tables

**Figure 1 vetsci-12-00539-f001:**
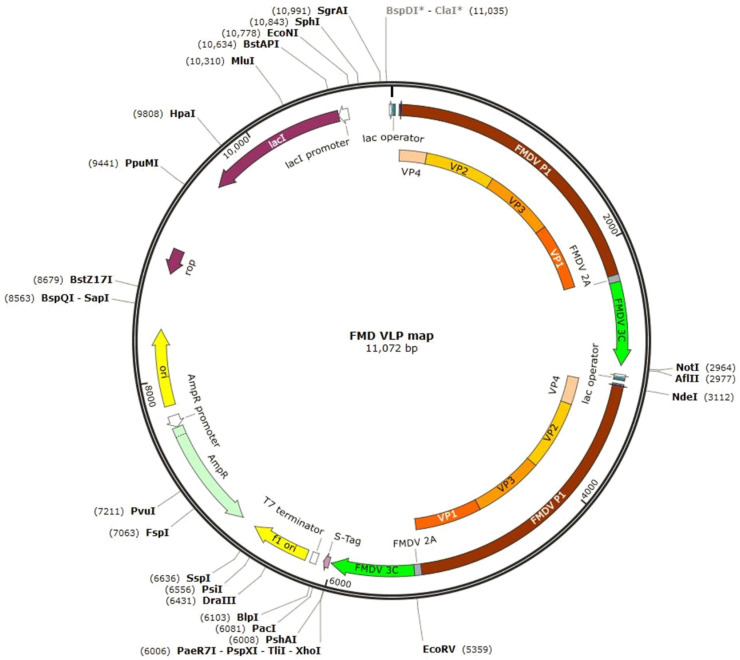
Recombinant plasmid map showing FMD VLP structural proteins. The asterisk (*) indicates a restriction site blocked by Dam methylation, which prevents enzyme recognition or cleavage at that location. To modify the methylation status, use the “Change Methylation” option in the Edit menu.

**Figure 2 vetsci-12-00539-f002:**
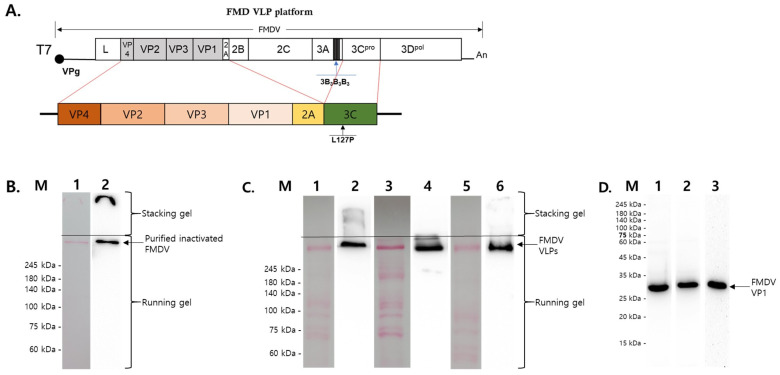
Schematic diagram and Western blot of recombinant FMD virus-like particles. (**A**) Schematic diagram of the recombinant gene expressing the structural proteins of FMD VLPs. (**B**) Western blot of purified inactivated A-type FMDV in non-denatured SDS-PGAE. M: Marker, lane 1: ponceau staining lane of purified inactivated A YC FMDV. Lane 2: Western blot of purified inactivated A YC FMDV. (**C**) Western blots of purified samples of each type of FMDV VLP in non-denatured SDS-PGAE. Lanes 1, 3, and 5: ponceau staining of purified VLPs of O PA2, A YC, and Asia1 Shamir; lanes 2, 4, and 6: Western blots of purified VLPs of O PA2, A YC, and Asia1 Shamir. (**D**) Western blot analysis of purified FMDV VLPs under reducing SDS-PAGE conditions. M: Marker, lanes 1–3: Western blot of each purified VLP of O PA2, A YC, and Asia1 Shamir.

**Figure 3 vetsci-12-00539-f003:**

Structural Characterization of purified FMDV VLPs. (**A**–**C**) Transmission electron microscopy analysis of each VLP for O PA2, A YC, and Asia1 Shamir. A small number of rod-shaped particles (indicated by arrows) were also observed, which are presumed to be non-specific protein aggregates or assembly intermediates (12S or 5S forms) commonly seen in partially purified *E. coli*-derived VLP preparations. (**D**) Sucrose gradient fractionation analysis of concentrated PEG-precipitated O PA2 VLP. (**E**) Western blot analysis of sucrose gradient fraction samples of FMDV VLP for O-type, M: Marker, lanes 1~5: Fractionation samples of numbers 6~10. (**F**) HPLC analysis of sucrose fractionated 75S VLP of O-type and FMD virion of O-type.

**Figure 4 vetsci-12-00539-f004:**
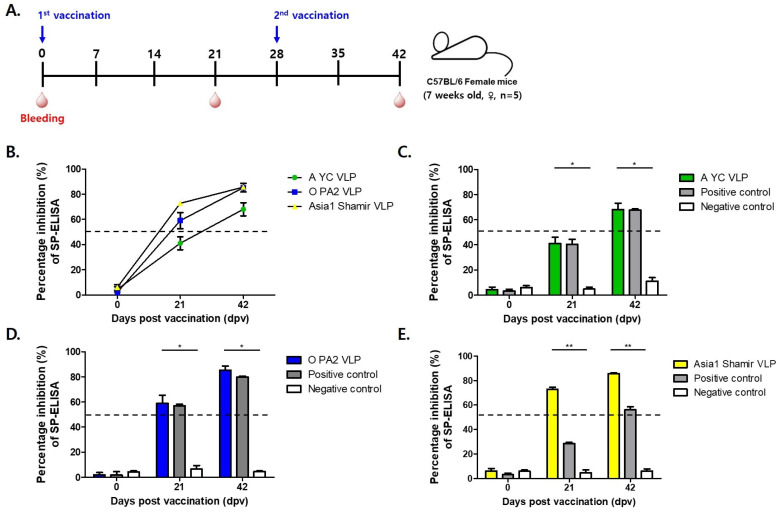
Immunogenicity of mice vaccinated with recombinant FMD VLP antigens. (**A**) C57BL/6 mice (n = 5/group) were administered two doses of the test vaccine containing recombinant FMD VLP (A YC VLP, O PA2 VLP, Asia1 Shamir VLP) antigens at 28 dpv intervals. (**B**) Time-course changes in percent inhibition (PI) values from SP-ELISA at 0, 21, and 42 days post-vaccination (dpv) for each VLP-immunized group (O, A, and Asia1). (**C**–**E**) Percent inhibition (PI) values at Day 0, 21, and 42 post-vaccination in each VLP-immunized group compared with positive (inactivated FMDV vaccine; Boehringer Ingelheim) and negative (antigen-free control group) controls for A-type (**C**), O-type (**D**), and Asia1-type (**E**) FMDV. Data are mean ± SEM. * *p* < 0.1, ** *p* < 0.01 (Kruskal–Wallis test followed by Dunn’s multiple comparisons tests).

**Figure 5 vetsci-12-00539-f005:**
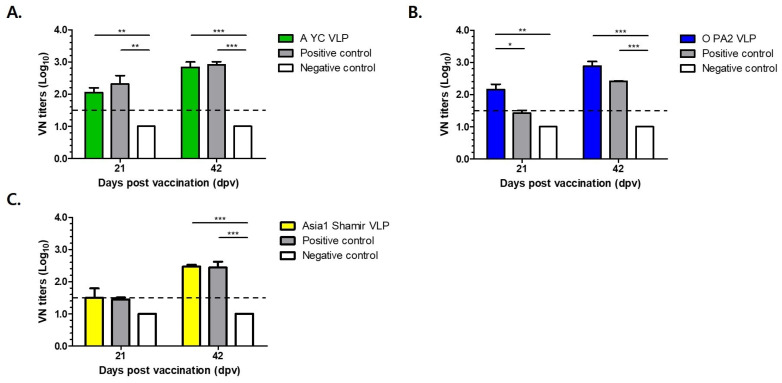
Virus neutralization (VN) titers in mice vaccinated with recombinant FMD VLP antigens. (**A**) VN titers of FMDV A strains in immunized mice. (**B**) VN titers of FMDV O strains in immunized mice. (**C**) VN titers of FMDV Asia1 strains in immunized mice. The dotted line represents a neutralizing antibody titer of 1:32. Positive control: Boehringer Ingelheim inactivated FMDV vaccine group. Negative control: Antigen-free control group. Data are mean ± SEM. * *p* < 0.1, ** *p* < 0.01, *** *p* < 0.001 (Kruskal–Wallis test followed by Dunn’s multiple comparisons tests).

**Figure 6 vetsci-12-00539-f006:**
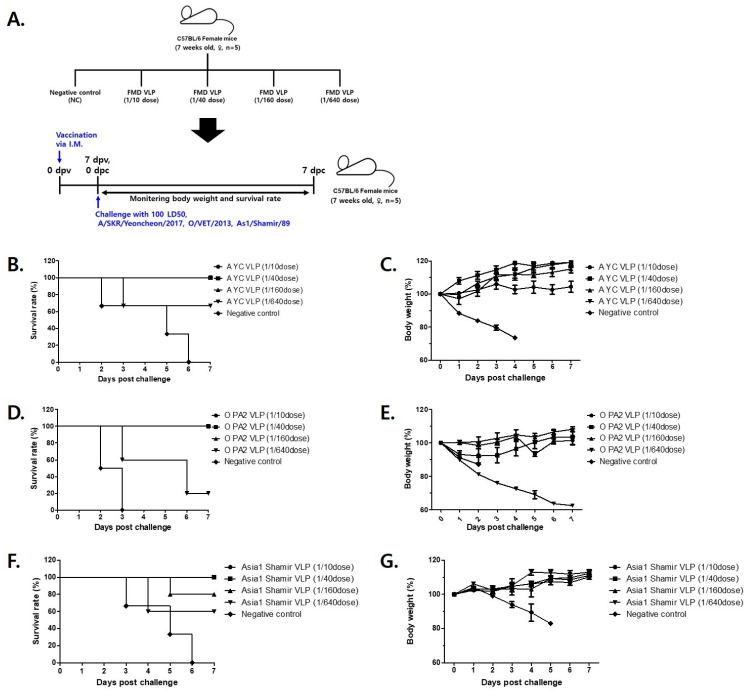
Survival and body weight of immunized mice challenged with FMD virus 7 days after vaccination with FMD VLPs. (**A**) C57BL/6 mice (n = 5/group) were administered a test vaccine containing recombinant FMD VLP vaccine at 1/10, 1/40, 1/160, 1/640 doses of each FMDV serotype (A YC VLP, O PA2 VLP, Asia1 Shamir VLP) and then infected with FMDV. (**B**,**D**,**F**) Survival rates post-challenge with A/SKR/Yeoncheon/2017 (**B**), O/VET/2013 (**D**) or As1/Shamir/89 (**F**). (**C**,**E**,**G**) Changes in body weight post-challenge with A/SKR/Yeoncheon/2017 (**C**), or O/VET/2013 (**E**) or As1/Shamir/89 (**G**). The data represent the mean ± SEM of triplicate measurements (n = 5/group).

**Figure 7 vetsci-12-00539-f007:**
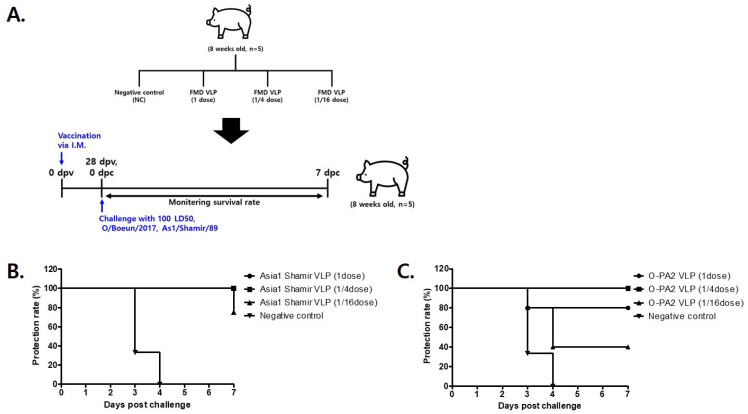
Survival of immunized swine challenged with FMD virus 28 days after vaccination with FMD VLPs. (**A**) Swine (n = 5/group) were administered a test vaccine containing recombinant FMD VLP antigens (O PA2 VLP, Asia1 Shamir VLP) and then infected with FMDV. (**B**,**C**) Survival rates post-challenge with O/Boeun/2017 (**B**) or As1/Shamir/89 (**C**).

**Table 1 vetsci-12-00539-t001:** Humoral immune responses induced by FMDV VLP vaccines in immunized mice.

Serotype	SP-ELISA PI (%)		VNT (1:X)	
	21 dpv	42 dpv	21 dpv	42 dpv
**A-type**	<50%	>50%	112	676
**O-type**	>50%	>50%	143	767
**Asia1-type**	>50%	>50%	47	295

**Table 2 vetsci-12-00539-t002:** Protective efficacy (PD_50_) of FMDV VLP vaccines in mice and swine models.

Serotype	PD_50_	
	Mice	Swine
A-type	73.5	–
O-type	32.0	10.6
Asia1-type	55.7	22.6

## Data Availability

Data are available upon reasonable request from the corresponding author.

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
