# Peer review of "Foot-and-Mouth Disease Virus-like Particles Produced in E. coli as Potential Antigens for a Novel Vaccine"

_vetsci, 2025, doi:10.3390/vetsci12060539_

Round 1
Reviewer 1 Report
Comments and Suggestions for Authors
The overall content details need to be supplemented, and the details are in the PDF

Reviewer 2 Report
Comments and Suggestions for Authors
The manuscript discusses a novel approach for production of vaccine against FMD.
I have detected some issues and these are outlines below.
Introduction. Please explain in detail the precise gaps in the literature that would be filled through this work.
Some passages from the Introduction (for example, the 2nd paragraph) can be deleted, without the text losing any important context.
M&M. There is a serious omission about the lack of description for controls. Please add a new section, to describe in detail all controls (strains, animals, consumables, procedures) used in this study.
Table 1: please move to appendix.
Question to please address: How the kinetics of FMD virus replication can affect the protocol employed by the authors in the present study?
Use of ANOVA in analysis should have been preceded by demonstration of normality of the data. As this was not done, please employ non-parametric techniques.
Results. Excellent figures, well done.
Omission of tables is a concern. Please add tables to summarise the findings and to make reading of the results easier.
Discussion. The Discussion does not cover fully all the findings of the study. Please extend by adding further ideas, especially please address the issue of replication kinetics indicated above.
I expect at least 70 references for such a complex manuscript with a variety of ideas, hence please add in the revised manuscript.
The Discussion must be divided into two sub-sections to allow easier flow of reading.
Figure S1 please move into main text.
Conclusions. This is very OK.
Overall. The manuscript needs a significant revision and improvement. After resubmission, it should go again for peer review.
Recommendation. Major revision.
Round 2
Reviewer 2 Report
Comments and Suggestions for Authors
All issues were addressed. No further comments.